# Investigating the use of ultrasonography for the antenatal diagnosis of structural congenital anomalies in low-income and middle-income countries: systematic review protocol

Stephanie Michele Goley,[1] Sidonie Sakula-Barry,[2] Ann Kelly,[1] Naomi Wright[3]

► Additional material is published online only. To view, please visit the journal online (http://dx.doi.org/10.1136/bmjpo-2019-000538).

¹Department of Global Health & Social Medicine, King's College London, UK
²World Cancer Research Fund, London, UK
³King's Centre for Global Health and Health Partnerships, King's College, London, UK

**Correspondence to**
Stephanie Michele Goley;
stephanie.goley@kcl.ac.uk

## ABSTRACT

**Introduction** Congenital anomalies are the fifth leading cause of mortality in children under 5 years globally. The greatest burden is faced by those in developing countries, where over 95% of deaths occur. Many of these deaths may have been preventable through antenatal diagnosis and early intervention. This study aims to conduct a systematic review that investigates the use of antenatal ultrasound to diagnose congenital anomalies and improve the health outcomes of infants in low-income and middle-income countries (LMICs).

**Methods and analysis** A systematic literature review will be conducted using three search strings: (1) structural congenital anomalies, (2) LMICs and (3) antenatal diagnosis. Four electronic databases will be searched: MEDLINE, Embase, PubMed and the Cochrane Library. Title, abstract and full-text screening will be undertaken in duplicate by two reviewers independently using Covidence. Consensus will be sought from the wider authorship for discrepancies. Data extraction will be undertaken by the principal investigator. The primary analysis will focus on the availability and effectiveness of antenatal ultrasound for structural congenital anomalies. Secondary outcomes will include neonatal morbidity and mortality, termination rates and referral rates for further antenatal care. Descriptive statistics and a narrative synthesis will be included in the final report. The methodological quality of the included studies will be evaluated using the Cochrane-approved Risk of Bias for Non-Randomised Studies of Intervention and Risk of Bias in Randomised Trials V.2.0 tools.

**Ethics and dissemination** Ethical approval is not required for conducting the systematic review as there will be no direct collection of data from individuals. The results will be submitted for publication in a scientific journal and presented internationally.

**Conclusion** This is the first study, to our knowledge, to systematically review current literature on the use of antenatal ultrasound for the detection of congenital anomalies in LMICs. This is vital to define current practice, highlight global disparities and evaluate effects on health outcomes for infants in low-resource settings.

**PROSPERO registration number** CRD42019105620.

## What is already known on this topic?

► Ninety-seven per cent of deaths from congenital anomalies occur in developing countries. Many of these deaths could be prevented with early diagnosis and intervention.
► Ultrasound machines are widely accessible and commonplace in high-income countries, but a number of factors limit the accessibility and effectiveness of ultrasound in LMICs.

## What this study hopes to add?

► To systematically investigate, for the first time, the availability and effectiveness of antenatal ultrasound in low-income and middle-income countries (LMICs) in order to elucidate disparities in congenital anomaly detection.
► To evaluate the effects of antenatal ultrasound on the morbidity and mortality rates of neonates with a structural congenital anomaly in LMICs.

## INTRODUCTION

Congenital anomalies represent 9% of the global burden of surgical disease and are one of the leading causes of infant morbidity and mortality globally.[1–4] The burden of congenital anomalies is much greater in low-income and middle-income countries (LMICs), comprising over 95% of deaths from these conditions.[5 6] According to a report by the WHO, every year, approximately 303 000 infants die within 4 weeks of birth due to congenital anomalies.[7] However, this is likely an underestimation, as statistics are undoubtedly skewed from under-reporting, a lack of congenital anomaly registries and unreliable medical records in many LMICs.

Structural congenital anomalies are physical deformities that occur during intrauterine development.[7] These anomalies include, but are not limited to, gastrointestinal malformations, cleft lip and palate, heart defects, musculoskeletal anomalies and neural tube defects.[8] If detected early and provided appropriate surgical intervention following birth, many structural congenital anomalies can be corrected, avoiding preventable death or disability.[7]

Advancements in ultrasound technology have allowed for early detection of structural congenital anomalies, which allows mothers and physicians to plan for an appropriate place of delivery and surgical intervention after birth.[9] In many cases, this can result in a significantly reduced risk of morbidity and mortality.[10] While ultrasound machines are widely accessible and commonplace in high-income countries (HICs), a number of factors limit the access and effectiveness of antenatal ultrasound in LMICs.

This study aimed to systematically investigate, for the first time, the availability and effectiveness of antenatal ultrasound in the diagnosis of structural congenital anomalies in LMICs. Furthermore, it aimed to evaluate the effects of antenatal ultrasound on mortality and morbidity for neonates with a structural congenital anomaly in LMICs. Such information is vital to help clarify the existing disparities in antenatal ultrasound provision and the potential benefits for improved health outcomes through the delivery of this service in low-resource settings.

## METHODS
Preferred reporting items for systematic reviews and meta-analyses protocols guidelines will be followed in conducting this systematic review[11 12] (online supplementary file 1). If there are amendments to the protocol, they will be reported in the publication of the results.

### Aim
The objective of this study was to conduct a systematic review that investigates the use of antenatal ultrasound to diagnose congenital anomalies and to improve the health outcomes of infants in LMICs.

### Objectives
1. To systematically identify and describe studies that focus on the antenatal diagnosis of structural congenital anomalies in LMICs.
2. To evaluate the use and effectiveness of ultrasound machines in the antenatal diagnosis of structural congenital anomalies and to report the current practices and policies regarding congenital anomaly detection in LMICs.
3. To evaluate the effects of antenatal ultrasound on the morbidity and mortality rates of neonates with a structural congenital anomaly in LMICs.
4. To critique the methodological quality of the included articles.

### Patient and public involvement
Given that this is a systematic literature review, there will be no patient or public involvement for the data collection and review of the literature. However, public involvement will be important for prioritising antenatal ultrasound on the political agenda. It will also be crucial for improving current antenatal healthcare programmes. Following the conclusion of this study, international parent/patient support groups and charities involving structural congenital anomalies will be approached to assist with the dissemination of the findings via their websites, social media, in-person meetings and other appropriate routes. A plain English summary of the findings will be provided for this purpose.

### Search strategy
A search will be conducted using three search strings (table 1). Search string 1 will encompass structural congenital anomalies. Search string 2 will focus on the setting—LMICs. Finally, search string 3 will look at the antenatal diagnosis of structural congenital anomalies, particularly focusing on the use of ultrasound machines for detection. Boolean operators 'and' and 'or' will be used within the search to combine the search terms.

Further search will be conducted on the WHO website in order to get a more robust understanding of current programmes and policies in place for antenatal ultrasound. Researchers will initially search the following terms in the WHO Reproductive Health Library: ultrasound, ultrasonography, congenital anomalies, congenital abnormalities, congenital anomaly, congenital abnormality, birth defect, antenatal detection, prenatal detection, antenatal diagnosis and prenatal diagnosis. These terms were chosen based on an initial screening performed by the researchers, which suggested that these terms provided optimum sensitivity for procuring all relevant WHO literature for this review. Following the search of each term, the results will be expanded using a snowball strategy in order to ensure the inclusion of all relevant data. The terms that arise in the expanded search will be included in the final report.

### Published literature search
Using the Ovid program, an electronic database search will be conducted on MEDLINE, Embase, PubMed, and the Cochrane Library using the aforementioned search strategy. These searches will be filtered to only include studies with human subjects. An example of the search in MEDLINE can be found in online supplementary file 2.

### Inclusion/exclusion criteria
Only fetuses with a structural congenital anomaly as listed in search string 1 will be included. Studies focusing on non-structural congenital anomalies will be excluded. Furthermore, all studies must focus on participants from LMICs; research focusing on HICs will be excluded. Finally, studies that concentrate on antenatal diagnosis will be included, while studies focusing on postnatal

**Table 1** Search strings

| Search string 1 | Search string 2 | Search string 3 |
|---|---|---|
| Congenital Anomalies, Congenital Abnormalities, Congenital Malformation, Fetal Malformation, Birth Defects, Anencephaly, Conjoined Twins, Congenital Heart Defects, Anorectal Malformations, Anal Stenosis, Anal Atresia, Imperforate Anus, Biliary Atresia, Choledochal Cyst, Diaphragmatic Eventration, Esophageal Atresia, Tracheoesophageal Fistula, Intestinal Atresia, Duodenal Obstruction, Duodenal Atresia, Colonic Atresia, Malrotation, Apple Peel Syndrome, Congenital Diaphragmatic Hernias, Gastroschisis, Abdominal Wall Defects, Exomphalos, Omphalocele, Congenital Limb Deformities, Neural Tube Defects, Bronchogenic Cyst, Bronchopulmonary Sequestration, Congenital Cystic Adenomatoid Malformation of Lung, Renal Anomalies, Genito-urinary Anomalies, Maxillofacial Abnormalities, Mouth Abnormalities, Umbilical Hernia, Hirschsprung Disease, Ganglionic Megacolon, Rectosigmoid Aganglionosis, Colonic Aganglionosis, Intestinal Aganglionosis, Volvulus, Intestinal Volvulus | LMICs, Low- and Middle-Income Countries, Developing Countries, Low-Resource Settings, Underdeveloped Countries, Low-Income Countries, Middle-Income Countries, Limited Resource Settings, Africa South of the Sahara, Sub-Saharan Africa, Less Resourced Communities, Afghanistan, Albania, Algeria, American Samoa, Angola, Argentina, Armenia, Azerbaijan, Bangladesh, Belarus, Belize, Benin, Bhutan, Bolivia, Bosnia and Herzegovina, Botswana, Brazil, Bulgaria, Burkina Faso, Burundi, Cabo Verde, Cambodia, Cameroon, Central African Republic, Chad, China, Colombia, Comoros, Democratic Republic of the Congo, DRC, Republic of the Congo, Costa Rica, Cote d'Ivoire, Ivory Coast, Croatia, Cuba, Djibouti, Dominica, Dominican Republic, Ecuador, Egypt, El Salvador, Equatorial Guinea, Eritrea, Ethiopia, Fiji, Gabon, Gambia, Georgia, Ghana, Grenada, Guatemala, Guinea, Guinea-Bissau, Guyana, Haiti, Honduras, India, Indonesia, Islamic Republic of Iran, Iraq, Jamaica, Jordan, Kazakhstan, Kenya, Kiribati, Democratic People's Republic of Korea, Kosovo, Kyrgyz Republic, Lao PDR, Laos, Lebanon, Lesotho, Liberia, Libya, Macedonia Republic, Madagascar, Malawi, Malaysia, Maldives, Mali, Marshall Islands, Mauritania, Mauritius, Mexico, Micronesia, Moldova, Nauru, Nepal, Nicaragua, Niger, Nigeria, Pakistan, Panama, Papua New Guinea, Paraguay, Peru, Philippines, Romania, Russian Federation, Rwanda, Samoa, Sao Tome and Principe, Senegal, Serbia, Sierra Leone, Solomon Islands, Somalia, Somaliland, South Africa, South Sudan, Sri Lanka, Saint Lucia, Saint Vincent and the Grenadines, Sudan, Suriname, Swaziland, Syrian Arab Republic, Syria, Tajikistan, Tanzania, Thailand, Timor-Leste, East Timor, Togo, Tonga, Tunisia, Turkey, Turkmenistan, Tuvalu, Uganda, Ukraine, Uzbekistan, Vanuatu, Venezuela, Vietnam, West Bank and Gaza, Republic of Yemen, Zambia, Zimbabwe | Antenatal Diagnosis, Prenatal Diagnosis, Antenatal Screening, Prenatal Screening, Antenatal Ultrasound, Prenatal Ultrasound, Antenatal Ultrasonography, Prenatal Ultrasonography |

diagnosis will be excluded. Included studies will be limited to the English language.

## Study design

No filters will be applied to study types; thus, all forms of evidence-based research will be included. This includes, but is not limited to, systematic reviews and meta-analyses, randomised controlled trials, prospective and retrospective cohort studies, case–control studies and case series/reports. Qualitative studies and WHO/government policy documents and guidelines will be included to provide insight into the current practices regarding the provision of antenatal ultrasound and to help contextualise the use of antenatal ultrasound in LMICs.

## Methodological quality

To assess the methodological quality of the studies, the researchers will use the Cochrane Risk of Bias for Non-Randomised Studies of Intervention (ROBINS-I) and the revised tool to assess Risk of Bias in Randomised Trials (RoB) V.2.0.[13][14] These tools are widely used for critically appraising study methodology. For the purposes of this study, these tools will be used as a means of quality assessment, rather than as an inclusion/exclusion determinant. Two independent reviewers will score the articles according to the ROBINS-I and RoB V.2.0 criteria, and any discrepancies will be assessed by the research team until unanimity is reached.

## Study screening

References from the search results will be added to EndNote X8 and duplicates will be removed. The articles will then be uploaded to Covidence for the screening process. Two reviewers will screen the titles and abstracts in duplicate, removing any articles that do not meet the inclusion criteria. The remaining articles will be assessed by two reviewers independently in full text, and any remaining articles that do not adhere to the study criteria will be removed.

## Data extraction

Data extraction will be undertaken by the principal investigator. The following data will be extracted: place of study, study type, publication status, study population, patient cohort, gestational age at the time of diagnosis, type of anomaly, type of ultrasound, per cent receiving antenatal ultrasound, per cent with any anomaly detected, per cent with accurate antenatal diagnosis, reported sensitivity and specificity of antenatal ultrasound, training of ultrasound technician, referral rate for further antenatal care, referral rate to a tertiary paediatric surgery centre, termination rate, mortality rate and complications as per the Clavien-Dindo classification. The following data will be collected regarding antenatal ultrasound policies: date, country/region, governing body, population coverage, intervention(s) and outcomes as detailed previously.

## Data synthesis

Descriptive statistics and narrative synthesis will be used. The primary outcomes are the availability and

effectiveness of ultrasound for the antenatal diagnosis of structural congenital anomalies in LMICs. The secondary outcomes are the effects of antenatal ultrasound diagnosis on neonatal morbidity and mortality, termination rates and referral rates for further antenatal care. Data will be presented according to the following categories: availability of antenatal ultrasound, effectiveness of antenatal ultrasound, training of personnel performing the ultrasound examination, secondary outcomes and antenatal ultrasound policies in LMICs. Information relating to the provision of antenatal ultrasound in isolation or as part of a perinatal care programme will be noted, as will any details regarding private/government provisions of such services.

Based on the authors' experience, it is unlikely that a meta-analysis will be feasible due to limited availability of data. However, if there are appropriate data, a meta-analysis will be undertaken in duplicate by two independent authors and discrepancies will be resolved among the wider authorship. Appropriate data will be defined as two or more sufficiently homogeneous studies comparing morbidity and/or mortality between one group who has received antenatal ultrasound and another who has not. In addition, a meta-analysis may be undertaken if two or more studies have compared an intervention to improve antenatal ultrasound coverage or detection rates in a population. Meta-analysis will be undertaken in Stata 16 and results will be presented using a forest plot. If there are over 10 studies in the meta-analysis, a funnel plot will be undertaken to assess publication bias and a Galbraith plot will be used to investigate heterogeneity in effect sizes. The quality of evidence will be assessed following grading of recommendations, assessment, development and evaluations (GRADE) guidelines.

## Limitations

It is beyond the feasibility of this study to include articles in languages other than English. The researchers acknowledge this is a limitation. In order to aid interpretation of the systematic review findings, the number of studies included and excluded due to non-English language will be reported and depicted geographically in the final report.

Furthermore, it is important to note that antenatal ultrasound has further diagnostic capabilities, such as detecting abnormal growth or improper placental position. This systematic review will only focus on the detection of structural congenital anomalies; however, it may be prudent to consider other uses of antenatal ultrasound in further investigations.

## DISCUSSION

The discrepancy in mortality due to congenital anomalies between HICs and LMICs is quite substantial. For instance, the survival rate for infants with gastroschisis in HICs is above 95%, while in many LMICs, there are few survivors of the condition.[10] This study aims to

investigate the use of antenatal ultrasound to diagnose congenital anomalies in LMICs. Further, this study will examine the current policies and programmes in place for antenatal ultrasound. Gaining a better understanding of the current policies and practices that increase the antenatal diagnosis of structural congenital anomalies may help to determine the most effective standards of practice. Increasing the early diagnosis of structural congenital anomalies may help to reduce the morbidity and mortality rates of congenital anomalies in LMICs. Many complications that arise from certain structural congenital anomalies could be avoided if the condition is detected antenatally and steps are taken to ensure safe delivery, such as planning the birth at a tertiary health-care facility. This study may provide crucial information regarding the effects of antenatal ultrasound on morbidity and mortality rates from congenital anomalies in LMICs. This in turn can inform future studies aimed at improving availability and quality of antenatal ultrasound in LMICs to ultimately improve the health outcomes of infants born with these conditions.

### Ethics and dissemination

The researchers of this study will be conducting a secondary analysis. No new data will be collected and there will be no direct interaction with participants. Therefore, it is not necessary to acquire ethical approval prior to conducting this review. Following data collection and analysis, the results will be submitted to a peer-reviewed scientific journal for publication. The results will also be shared in conferences pertaining to infant health, global health and global surgery.

**Acknowledgements** We thank the library services at King's College London for help with the systematic review process.

**Contributors** NW conceived the idea for this study. All authors devised the study design. SMG drafted the protocol with significant contributions from NW and AK. SMG and SSB will perform the literature review and SMG will draft the results.

**Funding** NW receives funding from the Wellcome Trust with the funding refererence [203905/Z/16/Z].

**Disclaimer** Since this is a protocol, there is no supporting data.

**Competing interests** None declared.

**Patient consent for publication** Not required.

**Provenance and peer review** Not commissioned; externally peer reviewed.

**Data availability statement** There are no data in this work.

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
