## [Reviewer comments · BMJ Paediatrics Open]

ARTICLE DETAILS

TITLE (PROVISIONAL)	Investigating the Use of Ultrasonography for the Antenatal Diagnosis of Structural Congenital Anomalies in Low- and Middle-Income Countries: A Systematic Review Protocol
AUTHORS	Goley, Stephanie; Sakula-Barry, Sidonie; Kelly, Ann; Wright, Naomi

VERSION 1 – REVIEW

REVIEWER	Reviewer name: Andrea Solnes Miltenburg Institution and Country: University of Oslo, Norway Competing interests: I declare that I have no competing interests
REVIEW RETURNED	07-Jul-2019

GENERAL COMMENTS	This is a straightforward and clear protocol. It is well written and includes all the details. I do miss some more information regarding the study type, which the authors have described currently as including all forms. However, the objectives point to the main interest of focus on quantitative studies. Perhaps the authors could rephrase the first objective related to 'cataloging studies' and rather state 'to systematically describe studies (or interventions) that focus on... etc. And perhaps in the paragraph on study types be more explicit which types of studies are expected to be of use. And why the authors also would be interested in qualitative studies, or descriptive studies exploring availability of ultrasounds for detecting anomalies etc. Some details would be helpful. Also the authors have not stated their specific outcome interests.
---

REVIEWER	Reviewer name: Allegaert karel Institution and Country: ku leuven, Belgium and erasmusmc, rotterdam Competing interests: none
REVIEW RETURNED	31-Jul-2019

GENERAL COMMENTS	I have read your protocol with great interest and with a background of neonatal intensive care, including neonatal surgery aspects. Although I value the idea and the research question, this more feels as a somewhat naïve question, since I assume that the use of ultrasound to detect antenatal structural congenital anomalies as a token of structured perinatal care is a more likely scenario ? This becomes in my reading more relevant for the second part of the aim (to improve health outcome of infants in LMIC setting) suggest that ultrasound is indeed part of a clinical care program. Have the authors considered aspects like sensitivity and specificity of ultrasound diagnosis ? ultrasound detects more issues and is not limited to detecting congenital anomalies, but also growth (restriction, or overgrowth - diabetes mediates), or issues like placental position so that the introduction of an ultrasound program has additional effects, either positive or negative.
--

	Public involvement: I get the message that for the search that perhaps public involvement is not so relevant, but when considering issues like priorities, or uptake of ultrasound programs, training, and as part of a perinatal care program, prioritization will necessitate public involvement, right ? two additional specific suggestions do you consider to use a Snowball concept on the WHO website information, since this may unveil specific programs in LMIC setting Inclusion/exclusion criteria: infants ? or fetuses ?
--	--

VERSION 1 – AUTHOR RESPONSE

Reviewer 1

Comment: This is a straight forward and clear protocol. It is well written and includes all the details. However, the objectives point to the main interest of focus on quantitative studies. Perhaps the authors could rephrase the first objective related to 'cataloging studies' and rather state ' to systematically describe studies (or interventions) that focus on... etc.

Response: The first objective has been rephrased as per the suggestion.

Section of the Manuscript: Objectives

Comment: I do miss some more information regarding the study type, which the authors have described currently as including all forms. And perhaps in the paragraph on study types be more explicit which types of studies are expected to be of use. And why the authors also would be interested in qualitative studies, or descriptive studies exploring availability of ultrasounds for detecting anomalies etc. Some details would be helpful

Response: The study design section has been updated to include information regarding which study types we expect to be of use and to add further details about the inclusion of qualitative studies.

Section of the Manuscript: Study Design

Comment: Also the authors have not stated their specific

Response: The specific outcomes have been

Section of the Manuscript: Data Synthesis

Reviewer 2

Comment: I have read your protocol with great interest and with a background of neonatal intensive care, including neonatal surgery aspects. Although I value the idea and the research question, this more feels as a somewhat naïve question, since I assume that the use of ultrasound to detect antenatal structural congenital anomalies as a token of structured perinatal care is a more likely scenario ? This becomes in my reading more relevant for the second part of the aim (to improve health outcome of infants in LMIC setting) suggest that ultrasound is indeed part of a clinical care program.

Response: Thank you very much for this comment. From our experience, antenatal ultrasound is widely utilised in isolation amongst the private sector in LMICs alongside the government system, which may be more integrated into a perinatal care program. It will be important to detail relevant findings from the literature regarding such provision of antenatal ultrasound – we have added this to the protocol.

Section of the Manuscript: Data Synthesis

Comment: Have the authors considered aspects like sensitivity and specificity of ultrasound diagnosis ?

Response: These will be evaluated as part of the 'effectiveness of antenatal ultrasound' outcome. Details have been added to the data synthesis section.

Section of the Manuscript: Data Extraction

Comment: Ultrasound detects more issues and is not limited to detecting congenital anomalies, but also growth (restriction, or overgrowth - diabetes mediates), or issues like placental position so that the introduction of an ultrasound program has additional effects, either positive or negative.

Response: This is an important point – thank you. Unfortunately, it is beyond the feasibility and scope of this study to include antenatal diagnosis of other clinical factors within the search criteria. The intent is to focus on structural congenital anomalies, however further investigations into other antenatal factors would be an interesting area for future studies and we will report on such findings if they appear in the included studies. A note on this has been added to the limitations section.

Section of the Manuscript: Limitations

Comment: Public involvement: I get the message that for the search that perhaps public involvement is not so relevant, but when considering issues like priorities, or uptake of ultrasound programs, training, and as part of a perinatal care program, prioritization will necessitate public involvement, right?

Response: Information has been added to address the role of public engagement in prioritizing antenatal ultrasound.

Section of the Manuscript: Patient and Public Involvement

Comment: Do you consider to use a Snowball concept on the WHO website information, since this may unveil specific programs in LMIC setting

Response: A snowball strategy has been added to the WHO website search.

Section of the Manuscript: Search Strategy

Comment: Inclusion/exclusion criteria: infants ? or fetuses ?

Response: This has been edited to specify fetuses.

Section of the Manuscript: Inclusion/ Exclusion Criteria